# O₂VIS: Occupancy-aware Object Association for Temporally Consistent Video Instance Segmentation

## Abstract

In this paper, we present Occupancy-aware Object Association for Video Instance Segmentation (O₂VIS), a new framework crafted to improve long-term consistency in instance tracking. We introduce the Instance Occupancy Memory (IOM) that tracks global instance features and their occupancy status to effectively differentiate between recurring and new objects. It ensures consistent tracking and effective management of object identities across frames, enhancing the overall performance and reliability of the VIS process. Moreover, we propose a Decoupled Object Association (DOA) strategy that handles existing and newly appeared objects separately to optimally assign indices based on occupancy. This technique enhances the accuracy of object matching and ensures stable and consistent object alignment across frames, especially useful in dynamic settings where objects frequently appear and disappear. Extensive testing and an ablation study confirm the superiority of our method over traditional methods, establishing new standards in the VIS domain. Notably, our O₂VIS achieves the best AP scores on the YouTube-VIS benchmarks for 2019, 2021, and 2022, with results of *70.1*, *66.2*, and *54.0*, respectively. We provide our source code here.

## 1 Introduction

Video instance segmentation (VIS) is a complex task that requires segmenting, classifying, and tracking objects across video frames (Yang et al., 2019). Recent breakthroughs in this field have been significantly propelled by the adoption of query-based segmentation networks (Cheng et al., 2021b; 2022), which have notably enhanced the precision of instance segmentation. These networks utilize object queries to extract distinctive features for each object within a frame and then group pixels on the image feature map to delineate object regions. Building on the capabilities of these architectures, many of the latest VIS approaches (Huang et al., 2022; Heo et al., 2022; Wu et al., 2022b; Heo et al., 2023; Ying et al., 2023; Zhang et al., 2023a; Li et al., 2023a; Kim et al., 2024) focus on developing sophisticated methods for instance association, aiming to improve the accuracy and efficiency of tracking objects throughout a video.

To enhance object representation across frames, cross-frame contrastive learning has been explored (Wu et al., 2022b; Li et al., 2023b; Ying et al., 2023), though it often necessitates heuristic post-processing as illustrated in Fig. 1-(a). Addressing this, recent transformer-based trackers (Heo et al., 2023; Zhang et al., 2023a) offer directly aligned predictions without the need for post-processing, using cross-frame attention to dynamically reconstruct object representations from prior frames (Fig. 1-(b)). Despite their benefits, such methods typically depend on previous frame data, potentially affecting long-term tracking consistency. In both approaches, Some models enhance consistency using a dynamic memory updated via object similarity (Wu et al., 2022b; Heo et al., 2023; Ying et al., 2023) or momentum (Gao & Wang, 2023), yet challenges remain in consistently updating and associating object representations, as detailed in Sec. A.1, suggesting areas for further enhancement in VIS technologies.

To tackle the challenges inherent in VIS, we introduce *Occupancy-aware Object Association for Video Instance Segmentation (O₂VIS)*, a system designed for temporally consistent tracking. Central to our approach is an *Instance Occupancy Memory (IOM)*, which incorporates global instance

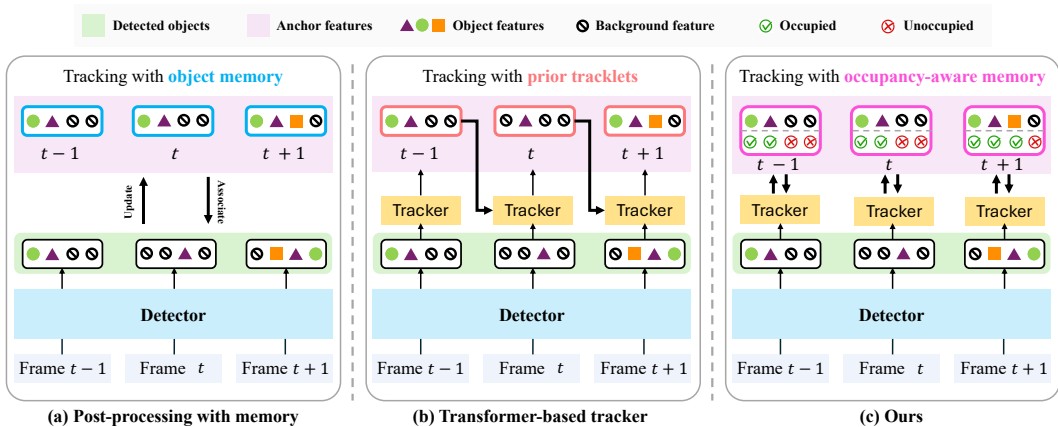

Figure 1: Comparison of VIS approaches. (a) The post-processing with memory method tracks objects by associating updated objects in memory with current ones, as in (Ying et al., 2023). (b) The transformer-based trackers associate previously tracked objects with current ones, as in (Zhang et al., 2023a). (c) Our method updates the memory using the results from the tracking network and occupancy, then utilizes this information for object association.

features and their occupancy status to indicate whether objects have previously appeared. The system meticulously updates an occupied index set, essential for accurately assigning indices to newly appearing objects while differentiating them from those already occupied. This design focuses on foreground objects, facilitating robust updates of object information. This feature is particularly beneficial in dynamic environments where objects may frequently appear and disappear. Such a strategy ensures consistent tracking and effective management of object identities across frames, enhancing the overall performance and reliability of the VIS process.

Our approach, $O_2$VIS, proposes a *Decoupled Object Association (DOA)* strategy that effectively utilizes global instance queries and their occupancy information for precise matching. This strategy separates the object association process into two phases. Initially, the method tracks the existing objects, ensuring their identities are persistently maintained across frames. It then aligns all instances in the current frame using an adaptive anchor query. To construct this query, we employ Hungarian matching technique, which strategically assigns newly appeared objects to unoccupied indices guided by the occupancy. This careful allocation ensures a more stable anchor query by correctly placing existing queries in occupied indices and new queries in unoccupied indices.

Our approach improves upon traditional cross-frame attention mechanisms, which frequently mis-link new objects to background indices, leading to learning conflicts. By distinctly separating the associations of existing and new objects, our decoupled strategy ensures precise and consistent object alignment across the video, avoiding common pitfalls of conventional methods. Extensive experiments demonstrate that our method surpasses traditional techniques, and an ablation study highlights the effectiveness of each technical contribution. Our contributions to the field are manifold and can be summarized as follows:

1. We present Occupancy-aware Object Association for Video Instance Segmentation, $O_2$VIS, a novel framework that leverages global instance queries for temporally consistent object matching.

2. We introduce the Instance Occupancy Memory (IOM), which stores object queries and their occupancy status, crucial for accurate object matching.

3. We introduce the Decoupled Object Association (DOA) that separates the object association step into matching existing objects in memory to queries and matching new objects.

4. Our model excels in challenging video environments, setting new benchmarks for state-of-the-art performance in video instance segmentation.

## 2 RELATED WORKS

**Video Instance Segmentation.** VIS primarily learns frame-to-frame feature associations using instance segmentation architectures. The seminal work, MaskTrack R-CNN (Yang et al., 2019), add a tracking head to Mask R-CNN (He et al., 2017), enhancing instance association. This is advanced by SipMask (Cao et al., 2020) and CrossVIS (Yang et al., 2021b), which improves temporal associations with cross-frame learning. Additionally, IDOL (Wu et al., 2022b) incorporates contrastive learning into a query-based architecture (Zhu et al., 2020), boosting online method performance. Beyond online methods, VisTR (Wang et al., 2021) apply DETR (Carion et al., 2020) for clip-level instance predictions, constrained by dense self-attention. Efficiency improvements are further pursued with IFC (Hwang et al., 2021), introducing a transformer with separate spatial and temporal attentions, and TeViT (Yang et al., 2022) and Seqformer (Wu et al., 2022a), which adapt vision transformer backbones to enhance temporal associations and video-level instance predictions, respectively. Recently, query-based segmentation networks have become fundamental to contemporary VIS approaches, prominently featuring Mask2Former (Cheng et al., 2022) as a key underlying technology. MinVIS (Huang et al., 2022) simplifies tracking by using post-processing based on cosine similarity between object features. VITA (Heo et al., 2022) enhances this by temporally associating frame-level queries to identify instance prototypes within a video. GenVIS (Heo et al., 2023) creates a tracking network that operates at the sub-clip level. CTVIS (Ying et al., 2023) employs contrastive learning across an expanded frame set to achieve detailed frame associations. DVIS (Zhang et al., 2023a) introduces a decoupled architecture that segments the processes into distinct tasks of segmentation, tracking, and refinement.

**Object Tracking with Memory.** Memory-based methods have shown significant advancements in video analysis, particularly in tasks requiring sustained long-term consistency, such as video object segmentation (Tokmakov et al., 2017; Xu et al., 2018; Duarte et al., 2019; Ventura et al., 2019; Huang et al., 2020; Zhang et al., 2020; Cheng & Schwing, 2022), video instance segmentation (Yang et al., 2019; Wu et al., 2022b; Heo et al., 2023; Ying et al., 2023), and video object tracking (Yang & Chan, 2018; Fu et al., 2021; Yan et al., 2021; Cai et al., 2022; Meinhardt et al., 2022; Zhao et al., 2023; Gao & Wang, 2023). Some research has successfully utilized external memory in multi-object tracking scenarios. MaskTrack R-CNN (Yang et al., 2019) employs an external memory to store predicted instance representations, updating them with new data from the latest frame through a straightforward replacement rule. Building on this, with the introduction of DETR (Carion et al., 2020), Meinhardt et al. (Meinhardt et al., 2022) developed TrackFormer, which innovates a *tracking-by-attention* approach. This method uses object query tokens to maintain temporal instance memories, enhancing tracking continuity. Recent advancements have further improved long-term consistency by either stacking these tokens in a memory buffer (Cai et al., 2022) or applying momentum to update the tokens (Gao & Wang, 2023; Heo et al., 2023). Despite these advancements, current techniques do not thoroughly address the challenges posed by the initial appearance or eventual disappearance of objects, often due to occlusion. This oversight suggests there remains significant potential for further enhancements in memory-based tracking methods.

## 3 PRELIMINARY

**Transformer-based Tracker.** Video instance segmentation (VIS) involves segmenting and tracking objects consistently across video frames. To address this challenge, recent studies (Huang et al., 2022; Heo et al., 2022; 2023; Ying et al., 2023; Li et al., 2023a; Zhang et al., 2023a;b) have adopted query-based segmentation network like Mask2Former (Cheng et al., 2022). The segmentation network $\mathcal{S}$ generates object representations $\tilde{Q}_t \in \mathbb{R}^{N \times C}$, categorical probabilities $P_t \in \mathbb{R}^{N \times (K+1)}$, and segmentation masks $M_t \in \mathbb{R}^{N \times H \times W}$ for each video frame $\{I_t\}_{t=1}^T$ as follows:

$$\left[ \tilde{Q}_t, P_t, M_t \right] = \mathcal{S}\left(I_t\right), \;\; \forall t = \{1, \dots, T\}, \tag{1}$$

where $N$ is a sufficiently large number to detect objects in an image, while $H$, $W$, and $C$ denote the height, width of predicted mask and the channel dimensions of the object representations, respectively. The categorical prediction head in the segmentation network classifies objects into $K$ categories or as no object $\varnothing$. The class label $c_t \in \mathbb{R}^N$ is determined by applying the argmax operation to the probability matrix $P_t$ along the $(K + 1)$ dimension.

The principal challenge in VIS is the maintenance of consistent object representations across frames, ensuring that the aligned sequences of predicted objects $Q_t \in \mathbb{R}^{N \times C}$ correspond to the same physical entities throughout the video as follows:

$$Q_t = \mathcal{T}\left(Q_{t-1}, \tilde{Q}_t\right), \ \forall t \in \{1, \dots, T\}, \tag{2}$$

where $\mathcal{T}$ is a tracking network consisting of multiple transformer blocks and the initial queries $Q_0$ are initialized as raw features using raw features extracted from the segmentation network, $\hat{Q}_1$, from the first frame. Recent state-of-the-art methods (Heo et al., 2023; Zhang et al., 2023a) use a transformer-based tracker that encodes object features from each frame, informed by the sequence from previous frames. Cross-attention layers within these transformers align current frame objects with previous ones based on feature similarity, ensuring accurate and consistent tracking across the video.

**Discussion.** The tracking network as outlined in Eq. (2) encounters two primary challenges. Firstly, the methodology relies on maintaining consistent object indices once they are established, but this consistency is not guaranteed. Given that object association hinges on comparisons with previous frame objects, an object that temporarily disappears and then reappears in the scene may be incorrectly indexed. Although strategies such as updating long-term memory with momentum (Gao & Wang, 2023) or enhancing object-wise similarity (Heo et al., 2023) have been proposed, they fail to preserve object identities reliably over time. The second issue concerns the incorporation of new objects into the scene. The cross-attention mechanism currently reconstructs objects in a frame based on their similarity to previously identified objects. Consequently, indices for new objects, which should ideally remain unassigned until needed, often default to background representation. This leads to a flawed learning process where new objects are mistakenly associated with background features, compromising the system's ability to accurately track new entries into the scene. To address these issues, we introduce an Occupancy-aware Object Association for Video Instance Segmentation, referred to as $O_2$VIS described in Sec. 4.

## 4  $O_2$VIS

As illustrated in Fig. 2, our model consists of two tracking networks, $\mathcal{T}_E$ for existing objects and $\mathcal{T}_A$ for all objects including new objects. These two networks align the object features produced by the pretrained segmentation network $\mathcal{S}$. Central to our architecture is the global instance occupancy memory, denoted as $\mathcal{M}$. Our approach incorporates two principal components to enhance tracking precision and consistency: an instance occupancy memory in Sec. 4.1 and a decoupled object association strategy in Sec. 4.2. These sections detail how they contribute to maintaining object continuity and seamlessly integrating new objects into the scene.

### 4.1  INSTANCE OCCUPANCY MEMORY

To enhance tracking in video instance segmentation (VIS), we introduce the global instance occupancy memory, a crucial component of our $O_2$VIS framework. This memory system at any given time $t$, denoted as $\mathcal{M}_t = \{\mathcal{O}_t, \mathcal{Q}_t\}$, consists of an occupancy indicators $\mathcal{O}_t \in \mathbb{B}^N$ and corresponding object features $\mathcal{Q}_t \in \mathbb{R}^{N \times C}$. This design accumulates the aligned queries $\dot{Q}_t \in \mathbb{R}^{N \times C}$ and the occupancy $O_t \in \mathbb{B}^N$ information over time as follows:

$$\mathcal{Q}_t = (1 - p_t)\mathcal{Q}_{t-1} + p_t \dot{Q}_t, \ \ \mathcal{O}_t = \mathcal{O}_{t-1} \vee O_t, \tag{3}$$

where $p_t$ is the foreground probability for each index calculated by summing the categorical probabilities $\{P_t^k\}_{k=1}^K$ for all classes excluding the no-object probability $P_t^{(K+1)}$. Our method is initialized at $\mathcal{Q}_0$ and $\mathcal{O}_0$ with $\dot{Q}_1$ and a zero vector $\mathbf{0}$ of length $N$, respectively. $O_t^n$ for each object $n$ is defined by:

$$O_t^n = \begin{cases} 0 & \text{if } c_t^n = \varnothing, \\ 1 & \text{otherwise.} \end{cases} \tag{4}$$

This dynamic update process allows the memory to adaptively reflect changes from the current video frame, thereby enabling more precise tracking and identification of objects as they appear, move, and potentially disappear within the scene. This memory design offers two main advantages critical

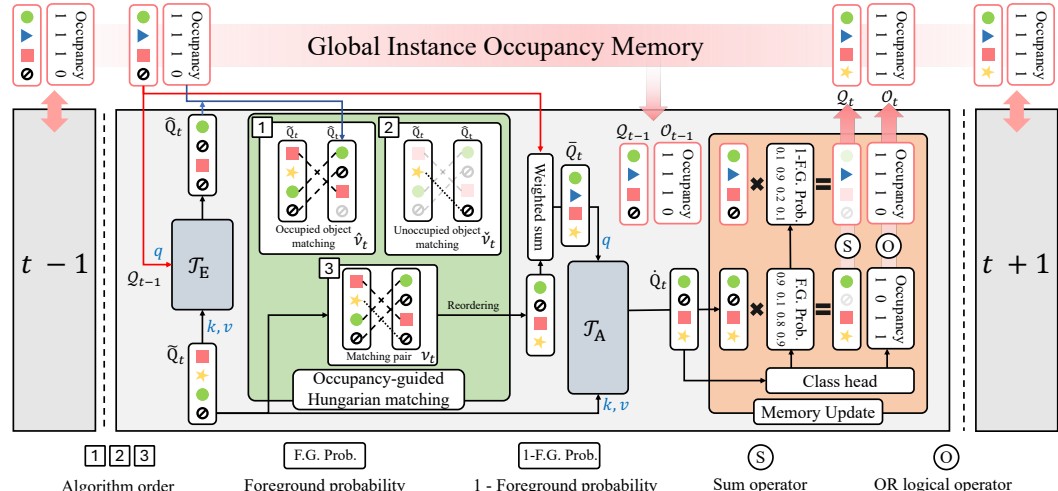

Figure 2: **Overall pipeline of our O₂VIS.** For each frame, the tracker $\mathcal{T}_E$ utilizes $\mathcal{Q}_{t-1}$ from the global instance occupancy memory and $\tilde{Q}_t$ from the segmentation network to predict aligned features for matched existing objects. Occupancy-guided Hungarian matching assigns newly appeared objects to unoccupied tokens, using the occupancy prior $\mathcal{O}_{t-1}$. The resulting $\bar{Q}_t$, a combination of rearranged $\hat{Q}_t$ and existing $\mathcal{Q}_{t-1}$, serves as the adaptive anchor for $\mathcal{T}_A$. Updates to $\mathcal{Q}_t$ and $\mathcal{O}_t$ are made based on foreground probability $p_t$ of objects in the current frame and occupancy information $O_t$, derived from the class head given the final instance queries $\dot{Q}_t$.

for effective video instance segmentation. Firstly, it allows for the strategic utilization of memory to ensure that newly appeared objects are not erroneously matched to indices already assigned to other objects. By continuously updating information about which indices are occupied and the specific objects associated with those indices, our system can adeptly assign new objects to suitable indices. Secondly, our approach enables consistent and robust maintenance of object information throughout the video sequence. Unlike traditional memory update methods that might prioritize background information, our method focuses on foreground objects. This prioritization is crucial in dynamic video environments where objects frequently appear and disappear, ensuring that our system maintains accurate and reliable object information over time.

## 4.2 Decoupled Object Association

Based on our observation described in Sec. 3, we design a decoupled object association strategy that consists of the following steps: 1) existing object tracking, 2) index assignment to new objects, and 3) object alignment with an adaptive anchor query. To avoid unnatural associations that can arise from directly tracking all objects observed in the instance queries $\tilde{Q}_t$, we initially focus on matching all instances to the existing objects recorded in the memory $\mathcal{Q}_{t-1}$ as follows:

$$\hat{Q}_t = \mathcal{T}_E \left( \mathcal{Q}_{t-1}, \tilde{Q}_t \right), \tag{5}$$

where $\hat{Q}_t$ represents aligned features of matched existing objects, and $\mathcal{T}_E$ denotes the existing object tracker. This mechanism specifically focuses on identifying existing objects in the current frame, thereby minimizing ambiguity in the object association process. It is important to note that $\hat{Q}_t$ excludes any objects that have newly appeared at time $t$. This ensures that the system accurately tracks and updates only those objects that persist across frames, without being confounded by newly introduced elements.

When new objects appear, they are assigned previously unallocated indices for accurate tracking. In this scenario, the queries from segmentation network $\tilde{Q}_t$ encompasses both existing and newly appeared objects, whereas the queries $\hat{Q}_t$ contains only the existing objects at occupied indices. By matching these two object features, appropriate indices for the new objects can be identified. To

facilitate this, we introduce an occupancy-guided Hungarian matching mechanism. This ensures that newly appeared objects are not erroneously matched to indices that are already occupied in the memory $\mathcal{O}_{t-1}$ from the previous time $(t-1)$. The matching starts by aligning objects corresponding to the occupied indices in $\hat{Q}_t$, followed by the remaining objects.

Given the entire index set $\mathcal{E}_N = \{1, \ldots, N\}$, we identify indices $\nu_t$, which can be separated into two subsets $\hat{\nu}_t$ and $\check{\nu}_t$, within the unaligned queries $\tilde{Q}$ as either occupied objects or unoccupied objects. Specifically, $\hat{\nu}_t$ denotes the ordered indices of occupied objects that correspond one-to-one with existing objects, while $\check{\nu}_t$ represents the ordered indices for the remaining unoccupied objects. The process of determining these indices is formalized as follows:

$$
\begin{aligned}
\hat{\nu}_t &= \arg\max_{\nu \subset \mathcal{E}_N} \sum_{n \in \hat{\mathcal{I}}} \text{sim}\left(\hat{Q}_t^n, \tilde{Q}_t^{\nu^n}\right), \quad \hat{\mathcal{I}} = \{n | O_{t-1}^n = 1\}, \\
\check{\nu}_t &= \arg\max_{\nu \subset \mathcal{E}_N \setminus \hat{\nu}_t} \sum_{n \in \check{\mathcal{I}}} \text{sim}\left(\hat{Q}_t^n, \tilde{Q}_t^{\nu^n}\right), \quad \check{\mathcal{I}} = \{n | O_{t-1}^n = 0\},
\end{aligned} \tag{6}
$$

where $\text{sim}(\cdot, \cdot)$ measures cosine similarity. To ensure robust object association, adaptive anchor queries $\bar{Q}_t$ is created by blending the current object queries $\tilde{Q}_t$ with existing object queries $\mathcal{Q}_{t-1}$ with the obtained matching relation $\hat{\nu}_t$ and $\check{\nu}_t$ rather than solely relying on the matched objects as query:

$$
\bar{Q}_t^n = \begin{cases} \text{Adp}(\tilde{Q}_t^{\hat{\nu}_t^n}, \mathcal{Q}_{t-1}^n) & \text{if } O_{t-1}^n = 1, \\ \text{Adp}(\tilde{Q}_t^{\check{\nu}_t^n}, \mathcal{Q}_{t-1}^n) & \text{otherwise,} \end{cases} \tag{7}
$$

$$
\text{where } \text{Adp}(A, B) = \text{sim}(A, B) \cdot A + \left(1 - \text{sim}(A, B)\right) \cdot B.
$$

The adaptive anchor queries $\bar{Q}_t$ are processed through the alignment network $\mathcal{T}_A$ to align with the current frame's object features $\tilde{Q}_t$, producing the final instance queries $\dot{Q}_t$ as follows:

$$
\dot{Q}_t = \mathcal{T}_A\left(\bar{Q}_t, \tilde{Q}_t\right). \tag{8}
$$

### 4.3 TRAINING

**Early training.** In the initial stages of training, the quality of representations, such as $\hat{Q}_t$, can be poor, making effective object matching as outlined in Eq. (6) challenging. To improve early training outcomes and provide a more stable foundation for learning, we adopt the following approach:

$$
\bar{Q}_t = \text{Adp}(\tilde{Q}_t^*, \mathcal{Q}_{t-1}), \quad \text{where } \tilde{Q}_t^* = \text{Hungarian}\left(\tilde{Q}_{t-1}^*, \tilde{Q}_t\right), \forall t \in \{1, \ldots, T\}, \tag{9}
$$

where "Hungarian" refers to the Hungarian matching algorithm (Kuhn, 1955), employed to enhance the initial alignment of object representations between frames. The initial queries $\tilde{Q}_0^*$ are initialized as raw features using raw features extracted from the segmentation network, $\tilde{Q}_1$, from the first frame.

The initial outputs from the tracking networks $\mathcal{T}_E$ and $\mathcal{T}_A$ often exhibit considerable noise, which can impede the accuracy of the tracking process. To mitigate this, we adopt the approach described in (Zhang et al., 2023a) for ground truth assignment, detailed in Sec. A.2. Specifically, predictions $\hat{y}$ from $\tilde{Q}_t^*$ are used for assigning ground truth via Hungarian matching. This method is strategically implemented during the first half of the total training iterations. This phased application allows the model to adapt incrementally to the task's complexity, enhancing the quality of the training representations as the process evolves. Such a staged training approach not only stabilizes the learning curve but also significantly improves alignment and tracking accuracy over time.

**Training loss.** In our approach, we utilize a comprehensive loss function aligned with those (Cheng et al., 2021a; Li et al., 2023a; Zhang et al., 2023a). This function incorporates categorical cross-entropy, binary cross-entropy, and dice losses, which are pivotal for effectively training our model. We specifically focus on optimizing the networks $\mathcal{T}_E$ and $\mathcal{T}_A$, while keeping other parameters static. We use both the actual ground truth $y_t$ and a modified version, $\mathring{y}_t^n$, which adapts the indices by assigning the no object label $\varnothing$ and zero mask where new objects appear. This modification ensures that $\mathring{y}_t$

Table 1: Comparison on YouTube-VIS validation sets. ∗ denotes offline methods.

| Method | Backbone | YouTube-VIS 2019 | | | | | YouTube-VIS 2021 | | | | | YouTube-VIS 2022 | | | | |
|---|---|---|---|---|---|---|---|---|---|---|---|---|---|---|---|---|
| | | AP | $AP_{50}$ | $AP_{75}$ | $AR_1$ | $AR_{10}$ | AP | $AP_{50}$ | $AP_{75}$ | $AR_1$ | $AR_{10}$ | AP | $AP_{50}$ | $AP_{75}$ | $AR_1$ | $AR_{10}$ |
| MinVIS (Huang et al., 2022) | R50 | 47.4 | 49.0 | 52.1 | 45.7 | 55.7 | 44.2 | 66.0 | 48.1 | 39.2 | 51.7 | 23.3 | 47.9 | 19.3 | 20.2 | 28.0 |
| VITA∗ (Heo et al., 2022) | R50 | 49.8 | 72.6 | 54.5 | 49.4 | 61.0 | 45.7 | 67.4 | 49.5 | 40.9 | 53.6 | 32.6 | 53.9 | 30.3 | 30.3 | 42.6 |
| GenVIS (Heo et al., 2023) | R50 | 50.0 | 71.5 | 54.6 | 49.5 | 59.7 | 47.1 | 67.5 | 51.5 | 41.6 | 54.7 | 37.5 | 61.6 | 41.5 | 32.6 | 42.2 |
| DVIS (Zhang et al., 2023a) | R50 | 51.2 | 73.8 | 57.1 | 47.2 | 59.3 | 46.4 | 68.4 | 49.6 | 39.7 | 53.5 | 31.6 | 52.5 | 37.0 | 30.1 | 36.3 |
| TCOVIS (Li et al., 2023a) | R50 | 52.3 | 73.5 | 57.6 | 49.8 | 60.2 | 49.5 | 71.2 | 53.8 | 41.3 | 55.9 | 38.6 | 59.4 | 41.6 | 32.8 | 46.7 |
| DVIS-DAQ (Zhou et al., 2024) | R50 | 55.2 | 78.7 | **61.9** | 50.6 | **63.7** | 50.4 | 72.4 | 55.0 | 41.8 | 57.6 | 34.6 | - | 35.5 | - | 41.1 |
| DVIS++ (Zhang et al., 2023b) | R50 | 55.5 | **80.2** | 60.1 | 51.1 | 62.6 | 50.0 | 72.2 | 54.5 | 42.8 | 58.4 | 37.2 | 57.4 | 40.7 | 31.8 | 44.6 |
| Ours | R50 | **55.7** | 79.8 | 61.4 | **51.3** | 62.7 | **50.7** | 72.9 | 56.9 | 43.7 | 58.4 | 41.1 | 62.4 | 46.2 | 35.8 | 47.5 |
| MinVIS (Huang et al., 2022) | Swin-L | 61.6 | 83.3 | 68.6 | 54.8 | 66.6 | 55.3 | 76.6 | 62.0 | 45.9 | 60.8 | 33.1 | 54.8 | 33.7 | 29.5 | 36.6 |
| VITA∗ (Heo et al., 2022) | Swin-L | 63.0 | 86.9 | 67.9 | 56.3 | 68.1 | 57.5 | 80.6 | 61.0 | 47.7 | 62.6 | 41.1 | 63.0 | 44.0 | 39.3 | 44.3 |
| DVIS (Zhang et al., 2023a) | Swin-L | 63.9 | 87.2 | 70.4 | 56.2 | 69.0 | 58.7 | 80.4 | 66.6 | 47.5 | 64.6 | 39.9 | 58.2 | 42.6 | 33.5 | 44.9 |
| GenVIS (Heo et al., 2023) | Swin-L | 64.0 | 84.9 | 68.3 | 56.1 | 69.4 | 59.6 | 80.9 | 65.8 | 48.7 | 65.0 | 45.1 | 69.1 | 47.3 | 39.8 | 48.5 |
| TCOVIS (Li et al., 2023a) | Swin-L | 64.1 | 86.6 | 69.5 | 55.8 | 69.0 | 61.3 | 82.9 | 68.0 | 48.6 | 65.1 | 51.0 | 73.0 | 53.5 | 41.7 | 56.5 |
| DVIS++ (Zhang et al., 2023b) | ViT-L | 67.7 | 88.8 | 75.3 | 57.9 | 73.7 | 62.3 | 82.7 | 70.2 | 49.5 | 68.0 | 37.5 | 53.7 | 39.4 | 32.4 | 43.5 |
| DVIS-DAQ (Zhou et al., 2024) | ViT-L | 68.3 | 88.5 | 76.1 | 58.0 | 73.5 | 62.4 | 83.6 | 70.8 | 49.1 | 68.0 | 42.0 | - | 43.0 | - | 48.4 |
| DVIS++∗ (Zhang et al., 2023b) | ViT-L | 68.3 | 90.3 | 76.1 | 57.8 | 73.4 | 63.9 | 86.7 | 71.5 | 48.8 | 69.5 | 50.9 | 75.7 | 52.8 | 40.6 | 55.8 |
| Ours | ViT-L | 69.1 | 89.3 | 76.5 | 58.1 | 73.5 | 65.0 | 86.0 | 72.7 | 49.6 | 69.1 | 48.2 | 70.5 | 53.2 | 40.7 | 52.6 |
| Ours∗ | ViT-L | **70.1** | **90.7** | **77.7** | **58.5** | **74.8** | **66.2** | **88.6** | **74.9** | **49.9** | **70.6** | **54.0** | **77.1** | **57.9** | **43.2** | **58.8** |

reflects only predictions for existing objects, thereby enhancing the relevance and accuracy of the training process. The structure of our primary training loss is computed as follows:

$$\mathcal{L}_{\text{Track}} = \sum_{t=1}^{T} \sum_{n=1}^{N_{GT}} \left( \mathcal{L}\left( \mathring{y}_t^n, \hat{y}_t^{\acute{\sigma}(n)} \right) + \mathcal{L}\left( y_t^n, \dot{y}_t^{\acute{\sigma}(n)} \right) \right), \quad \acute{\sigma} = \arg\min_{\sigma \in \mathfrak{S}_N} \sum_{n=1}^{N_{GT}} \mathcal{L}_{\text{Match}}\left( y_{f(n)}^n, \dot{y}_{f(n)}^{\sigma(n)} \right), \tag{10}$$

where $\dot{y}_t$ and $\hat{y}_t$ are the predictions from the feature representations $\dot{Q}_t$ and $\hat{Q}_t$, respectively. $\mathfrak{S}_N$ represents a permutation of $N$ elements, and $\mathcal{L}_{\text{Match}}$ denotes a pair-wise matching cost (Cheng et al., 2021a). $f(n)$ is computed specifically for the frame in which each $n$-th object first appears, following the approach used in (Zhang et al., 2023a). This ensures that our model specifically learns from relevant, existing object features and avoids any confusion from non-object areas or noise.

To ensure that occupied indices in $\tilde{Q}$ consistently represent the same objects across frames and that unoccupied indices reflect changes, we implement a similarity-based loss function as follows:

$$\mathcal{L}_{\text{Sim}} = \frac{1}{T \cdot N_{GT}} \sum_{t=2}^{T} \sum_{n=1}^{N_{GT}} \left( \text{sim}\left( \tilde{Q}_t^{\nu_t^{\acute{\sigma}(n)}}, \mathcal{Q}_{t-1}^{\acute{\sigma}(n)} \right) - O_{t-1}^{\acute{\sigma}(n)} \right)^2. \tag{11}$$

Our model is jointly trained using an objective function that combines the tracking loss and the similarity loss, with a balance determined by the weight $\lambda_{\text{Sim}}$:

$$\mathcal{L}_{\text{Total}} = \mathcal{L}_{\text{Track}} + \lambda_{\text{Sim}} \mathcal{L}_{\text{Sim}}. \tag{12}$$

## 5 EXPERIMENTS

### 5.1 IMPLEMENTATION DETAILS

We evaluate the performance of our $O_2$VIS using standard benchmark datasets: YouTubeVIS datasets (2019, 2021, 2022) (Yang et al., 2019) and OVIS (Qi et al., 2022). For our segmentation network, we employ the Mask2Former architecture (Cheng et al., 2022) equipped with three distinct backbone encoders: ResNet-50 (He et al., 2016), ViT-L and ViT-H (Dosovitskiy et al., 2021). All backbones are initialized with parameters pre-trained on COCO (Lin et al., 2014). Our tracking framework integrates two networks $\mathcal{T}_E$ and $\mathcal{T}_A$, each comprising three transformer blocks and enhanced with a referring cross-attention layer (Zhang et al., 2023a) for improved accuracy. Our tracking networks are trained with all other parameters frozen as previous studies (Zhang et al., 2023a; Li et al., 2023a). We empirically set $\lambda_{\text{sim}}$ as 1.0. Further details are described in Sec. A.2.

### 5.2 MAIN RESULTS

Following the standard evaluation metrics, Average Precision (AP) and Average Recall (AR), we benchmark the performance of $O_2$VIS against the current state-of-the-art methods in video instance segmentation.

Table 2: Comparisons with state-of-the-art methods on OVIS validation sets.

| Method | Backbone | AP | AP$_{50}$ | AP$_{75}$ | AR$_1$ | AR$_{10}$ |
|---|---|---|---|---|---|---|
| VITA(Heo et al., 2022) | Swin-L | 27.7 | 51.9 | 24.9 | 14.9 | 33.0 |
| MinVIS(Huang et al., 2022) | Swin-L | 39.4 | 61.5 | 41.3 | 18.1 | 43.3 |
| IDOL(Wu et al., 2022b) | Swin-L | 40.0 | 63.1 | 40.5 | 17.6 | 46.4 |
| MDQE(Li et al., 2023b) | Swin-L | 41.0 | 67.9 | 42.7 | 18.3 | 45.2 |
| NOVIS(Meinhardt et al., 2023) | Swin-L | 43.0 | 66.9 | 44.5 | 18.9 | 46.3 |
| GenVIS(Heo et al., 2023) | Swin-L | 45.2 | 69.1 | 48.4 | 19.1 | 48.6 |
| DVIS(Zhang et al., 2023a) | Swin-L | 45.9 | 71.1 | 48.3 | 18.5 | 51.5 |
| TCOVIS(Li et al., 2023a) | Swin-L | 46.7 | 70.9 | 49.5 | 19.1 | 50.8 |
| CTVIS(Ying et al., 2023) | Swin-L | 46.9 | 71.5 | 47.5 | 19.1 | 52.1 |
| DVIS++(Zhang et al., 2023b) | ViT-L | 49.6 | 72.5 | 55.0 | 20.8 | 54.6 |
| Ours | ViT-L | 51.7 | 73.9 | **57.5** | **21.1** | 56.2 |
| UNINEXT(Yan et al., 2023) | ViT-H | 49.0 | 72.5 | 52.2 | - | - |
| Ours | ViT-H | **52.9** | **76.1** | 55.3 | 20.1 | **57.4** |

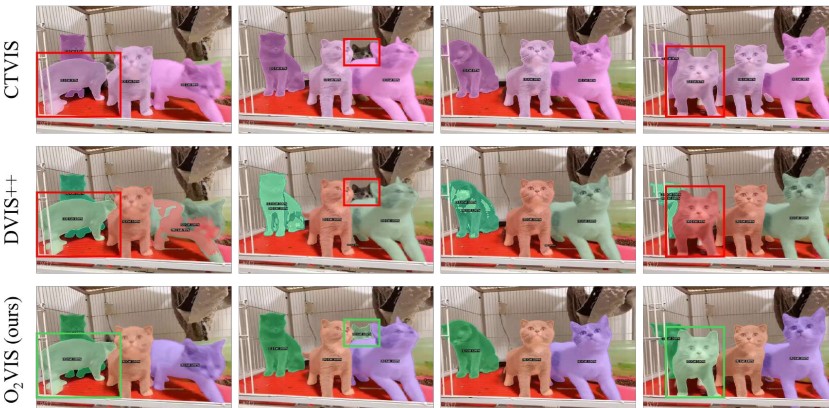

Figure 3: Qualitative comparison of O$_2$VIS with DVIS++ and CTVIS.

**Results on Youtube-VIS.** We compare O$_2$VIS with leading methods on the YouTube-VIS (YTVIS) datasets. The performance metrics, detailed in Tab. 1, show that O$_2$VIS outperforms state-of-the-art method, DVIS++, by achieving higher AP scores: +1.8 AP on YTVIS19, +2.2 AP on YTVIS21, and +3.1 AP on YTVIS22. Notably, O$_2$VIS also surpasses the performance of DVIS-DAQ, which addresses newly emerging and disappearing objects, under the same online setting equipped with a ViT-L backbone, achieving margins of +0.8 AP, +2.6 AP, and +6.2 AP on YTVIS19, YTVIS21, and YTVIS22, respectively. This significant improvement, particularly on longer video sequences, underscores O$_2$VIS's ability to maintain long-term consistency effectively. This is largely attributed to its innovative occupancy-aware memory mechanism that adapts dynamically to complex video contexts.

**Results on OVIS.** Our results on the OVIS benchmark are detailed in Tab. 2, where O$_2$VIS exhibits remarkable superiority over existing models. Notably, both with the ViT-L and ViT-H backbone, our model sets a new state-of-the-art by outperforming DVIS++ by +2.1 AP and UNINEXT by +3.9 AP. The strong performance across these configurations, particularly in a dataset characterized by frequent occlusions and dynamic object appearances, highlights the efficacy of O$_2$VIS's occupancy-aware memory and object association techniques in maintaining accurate and consistent tracking under challenging conditions. As shown in Fig. 3, our model demonstrates a clear advantage over previous models in complex scenes, successfully predicting object trajectories even in the presence of severe occlusions and multiple interacting objects.

### 5.3 ABLATION STUDY

**Instance occupancy memory.** We demonstrate the effectiveness of our instance occupancy memory through the results in Tab. 3-(a)-(1-4), which show a notable improvement by +1.5, +2.9, +7.2, and +3.6 AP on YTVIS19, YTVIS21, YTVIS22, and OVIS, respectively, compared to the baseline (1). The previous memory systems (2, 3) hardly update recent object information, resulting in

Table 3: Ablation studies on each component of $O_2$VIS. HM and OHM denote standard/occupancy-guided Hungarian matching, respectively. All experiments are evaluated using the AP metric.

(a) IOM, DOA

| | Architecture | Memory type | Tracking by | YTVIS19 | YTVIS21 | YTVIS22 | OVIS |
|---|---|---|---|---|---|---|---|
| (1) | $\mathcal{S}$ | None | HM | 51.1 | 45.0 | 26.8 | 26.4 |
| (2) | $\mathcal{S}$ | Similarity | HM | 51.8 (+0.7) | 46.8 (+1.8) | 28.5 (+1.7) | 28.6 (+2.2) |
| (3) | $\mathcal{S}$ | Momentum | HM | 48.0 (-3.1) | 41.2 (-3.8) | 30.9 (+4.1) | 13.7 (-12.7) |
| (4) | $\mathcal{S}$ | IOM | HM | 52.6 (+1.5) | 47.9 (+2.9) | 34.0 (+7.2) | 30.0 (+3.6) |
| (5) | $\mathcal{S}$ | IOM | OHM | 53.0 (+1.9) | 48.4 (+3.4) | 35.1 (+8.3) | 31.4 (+5.0) |
| (6) | $\mathcal{S} + \mathcal{T}_E$ | IOM | OHM | 53.6 (+2.5) | 49.6 (+4.6) | 39.1 (+12.3) | 33.9 (+7.5) |
| (7) | $\mathcal{S} + \mathcal{T}_E + \mathcal{T}_A$ | IOM | $\mathcal{T}_A$ | **55.7** (+4.6) | **50.7** (+5.7) | **41.1** (+14.3) | **37.1** (+10.7) |

(b) Anchor query for $T_A$

| Anchor | YTVIS19 | YTVIS21 | YTVIS22 | OVIS |
|---|---|---|---|---|
| $\tilde{Q}_t^{\nu_t}$ | 53.9 | 49.7 | 39.3 | 35.1 |
| $\mathcal{Q}_{t-1}$ | 55.2 | 50.1 | 39.2 | 36.2 |
| $\bar{Q}_t$ | **55.7** | **50.7** | **41.1** | **37.1** |

(c) Early training

| Early training | YTVIS19 | YTVIS21 | YTVIS22 | OVIS |
|---|---|---|---|---|
| ✗ | 55.1 | 50.4 | 40.0 | 36.7 |
| ✓ | **55.7** | **50.7** | **41.1** | **37.1** |

suboptimal tracking performance. Further details are described in Sec. A.1.1. This underscores the effectiveness of our approach, confirming that updating memory with foreground probability significantly enhances the model's accuracy and reliability, especially in challenging scenarios where precise object association is essential.

**Decoupled object association.** Tab. 3-(a)-(4-7) demonstrates the effectiveness of the decoupled object association, which consists of two trackers, $T_E$ and $T_A$, and occupancy-guided Hungarian matching. While the use of occupancy information alone in object matching, as seen in the comparison between (4) and (5), results in only a slight improvement of +0.4 to +1.4 AP, the performance significantly increases when both occupancy and existing object information are used for tracking. Specifically, comparing (4) and (6) shows a +4.0 AP gain on YTVIS22, demonstrating the effectiveness of combining these two strategies. Although $T_A$ further boosts the performance, as shown in Tab. 3-(b), the effectiveness depends on the choice of queries. While using only the object information from the current frame $\tilde{Q}_t^{\nu_t}$ as the anchor query results in performance similar to (a)-(6), using the object information from previous frames $\mathcal{Q}_{t-1}$ as the anchor query leads to better frame-to-frame associations and improves performance. The highest performance is achieved when both the current and previous frame information are combined and used as the anchor query $\bar{Q}_t$. These results demonstrate the strength of the Decoupled Object Association (DOA) in consistently tracking both existing and newly appeared objects.

**Early training strategy.** Although $O_2$VIS demonstrates robust performance, the initial stages of training the tracker can be particularly challenging due to the scarcity of learned information. To address this, we adopt a strategy where training initially relies on the outputs from the pre-trained segmentation network $\mathcal{S}$ for the first half of the training iterations. This approach facilitates a more stable and informed beginning to the learning process. The effectiveness of this strategy compared to conventional methods is detailed in Tab. 3-(c), showcasing the advantages of integrating the pre-trained knowledge at the early stages.

## 6 CONCLUSION

In this paper, we present Occupancy-aware Object Association for Video Instance Segmentation ($O_2$VIS), a robust method designed to maintain temporal consistency in VIS. At the heart of $O_2$VIS is the Instance Occupancy Memory, which utilizes global instance features and their occupancy status to effectively discern and track objects over time. The memory is essential for correctly assigning indices to new objects and distinguishing them from existing ones. We also propose the Decoupled Object Association strategy, integrating global instance queries with occupancy information to ensure precise object matching. This approach divides the object association process into two phases: continuously tracking existing objects to preserve their identities and aligning all current frame instances via an adaptive anchor query. This ensures both stable and accurate tracking. Extensive experiments confirm that our $O_2$VIS achieves top performance across the most standardized VIS benchmarks.

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

# A APPENDIX

## A.1 MOTIVATION FOR $O_2$VIS DESIGN

In this section, we outline the motivation behind the design of $O_2$VIS, focusing on three key points: **(1) Instance Occupancy Memory (IOM)** – A comparison between the proposed memory mechanism and previous memory methods, highlighting specific issues that arise during updates. **(2) Occupancy-guided Hungarian Matching (OHM)** – The importance of prioritizing the tracking of existing objects first. **(3) Decoupled Object Association (DOA)** – Problems found within the cross-attention block in existing transformer-based trackers.

### A.1.1 INSTANCE OCCUPANCY MEMORY (IOM)

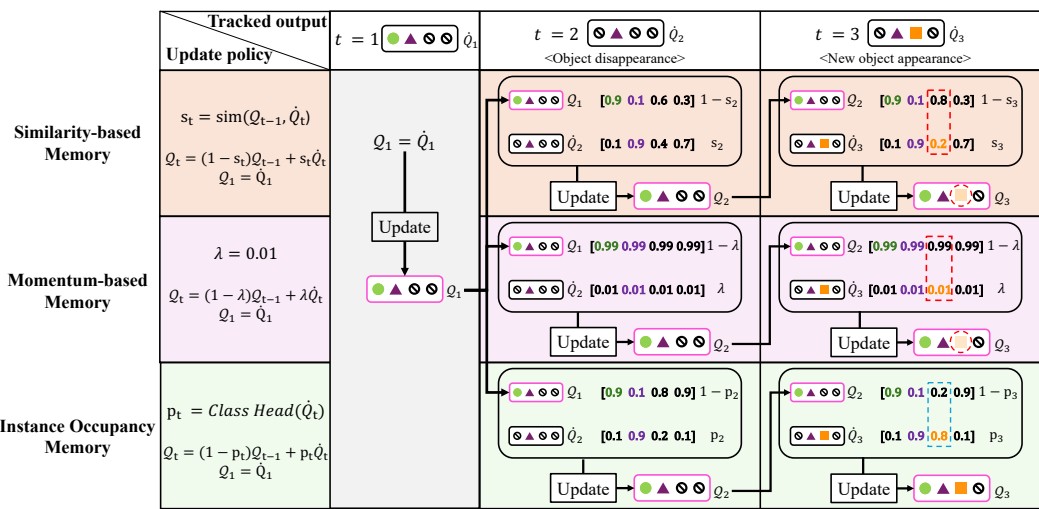

Figure 4: Comparison of three different memory mechanisms

Fig. 4 illustrates how the memory mechanisms operate, including similarity-based memory, momentum-based memory, and our proposed Instance Occupancy Memory (IOM).

**Similarity-based memory** (Wu et al., 2022b; Ying et al., 2023) updates the memory by assessing the similarity between the memory from time $t - 1$ and the objects at time $t$. When the object information at time $t$ shows high similarity to the memory objects, it receives a significant weight during the update; however, if the similarity is low, the update is minimal. This behavior is evident in Fig. 4, where at time $T = 2$, the memory retains information about the circle-shaped object that has disappeared, while the latest information for the triangle-shaped object is updated. As a result, we observe a performance improvement over the baseline in Tab. 3-(a)-(2). However, at time $T = 3$, when a new object appears, the memory structure struggles to update effectively due to the low similarity to the existing memory.

**Momentum-based memory** (Gao & Wang, 2023) assigns a high weight (*e.g.*, 99%) to the memory information from time $t - 1$ and a low weight (*e.g.*, 1%) to the objects at time $t$ during the update. As a result, the object information from the first frame is retained at a high ratio, while new objects or the latest information are hardly updated. Consequently, as shown in Tab. 3-(a)-(3), we observe a decline in performance across most datasets compared to the baseline.

**Instance-occupancy memory** uses the foreground probability of objects at time $t$ as a weight to update the object information. This means that the most recent and valid information is updated, allowing accurate memory updates even in scenarios where new objects appear, as seen in Fig. 4 at $T = 3$. Additionally, when objects disappear, the foreground probability is low, so the previous information is largely preserved, maintaining the object information well even in situations like $T = 2$. This novel mechanism ensures consistent memory updates, both for newly emerging and existing objects, offering robust performance in dynamic scenarios.

### A.1.2 OCCUPANCY-GUIDED HUNGARIAN MATCHING (OHM)

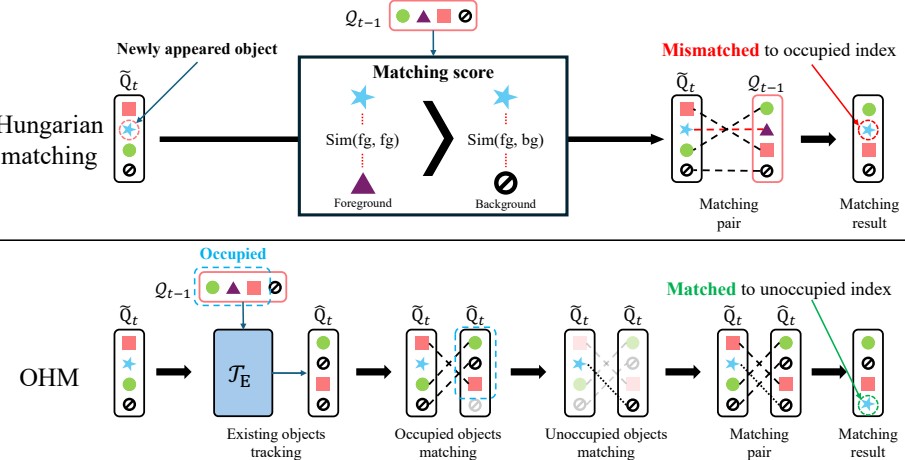

Figure 5: Hungarian matching vs OHM

Despite the introduction of IOM, there are still instances where the traditional Hungarian algorithm fails in accurate object matching. As shown in Fig. 4, when the model is unaware of occupied objects, newly appeared objects can be mistakenly assigned the index of existing objects due to cases where the similarity between foreground objects is higher than that between foreground and background objects. To prevent this, we first track only the objects common between the current frame and memory using $\mathcal{T}_E$, and then perform a two-step matching based on occupancy to ensure accurate object matching. Through this strategy, more accurate object matching is achieved in most scenarios, as demonstrated by the significant improvement in tracking performance shown in Tab. 3-(a), comparing (4) and (6).

### A.1.3 DECOUPLED OBJECT ASSOCIATION (DOA)

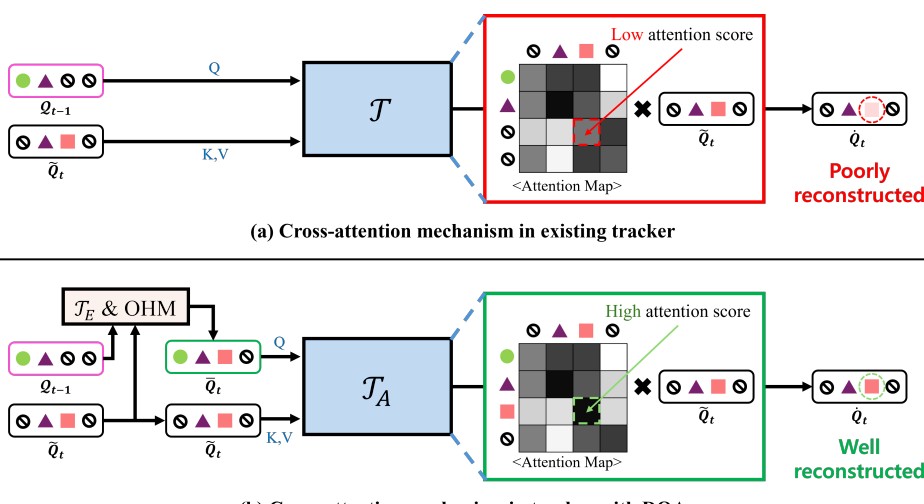

Figure 6: Illustration of cross-attentntion mechanism in transformer-based tracking network.

Transformer-based trackers (Heo et al., 2023; Zhang et al., 2023a) align the objects in the current frame with those in the previous frame by matching their order, with the cross-attention block being central to this operation. As illustrated in Fig. 6, the cross-attention block reconstructs the objects in the current frame based on the attention score between the objects in the previous frame and those in the current frame. However, to reconstruct new objects, the attention score between foreground object

features and background features must be high, which leads to unnatural learning and suboptimal tracking.

Based on this analysis, we propose generating an anchor query with decoupled strategy, which incorporates information from both previous and newly appeared objects through $\mathcal{T}_E$ and occupancy-guided Hungarian matching (OHM), and then leveraging it in the cross-attention. By providing appropriate anchors, our approach encourages the model to learn accurate tracking, particularly in scenarios with dynamic object appearances.

## A.2 TRAINING DETAILS

**Datasets.** We evaluate the performance of our $O_2$VIS using standard benchmark datasets: YouTube-VIS datasets (2019, 2021, 2022) (Yang et al., 2019) and OVIS (Qi et al., 2022), as detailed below. Introduced by (Yang et al., 2019) alongside the pioneering study on the Video Instance Segmentation (VIS) task, the YouTube-VIS datasets consist of high-resolution YouTube videos across 40 categories. The 2019 release includes 2,238 videos for training, 302 for validation, and 343 for testing. In its 2021 update (Yang et al., 2021a), the dataset was expanded to include 2,985 training videos, 421 validation videos, and 453 test videos, allowing for more extensive testing and development of VIS models. The 2022 version includes an additional 71 long videos in the validation set, while the training set remained the same as in the 2021 version. OVIS dataset (Qi et al., 2022) presents significant challenges with videos that often feature occlusions and long sequences that mirror complex real-world scenarios. This dataset is particularly demanding, with a greater number of objects and frames compared to YouTube-VIS, enhancing the difficulty of segmentation and tracking tasks. OVIS comprises 607 training videos, 140 validation videos, and 154 test videos, providing a robust platform for evaluating the effectiveness of VIS approaches under challenging conditions.

**Implementation Details.** For our segmentation network, we employ the Mask2Former architecture (Cheng et al., 2022) equipped with three distinct backbone encoders: ResNet-50 (He et al., 2016), ViT-L and ViT-H (Dosovitskiy et al., 2021). All backbones are initialized with parameters pre-trained on COCO (Lin et al., 2014). To improve memory efficiency with the ViT-L and ViT-H, we incorporate a memory-optimized VIT-Adapter (Chen et al., 2022), aligning with recent advancements in network efficiency (Zhang et al., 2023b). The segmentation network is further enhanced through pretraining with a contrastive learning approach for better object representation (Wu et al., 2022b; Ying et al., 2023; Zhang et al., 2023b; Lee et al., 2024). Our tracking framework integrates two networks $\mathcal{T}_E$ and $\mathcal{T}_A$, each comprising three transformer blocks and enhanced with a referring cross-attention layer (Zhang et al., 2023a) for improved accuracy.

For training, our tracking networks are trained with all other parameters frozen as previous studies (Zhang et al., 2023a; Li et al., 2023a). We employ the AdamW optimizer (Loshchilov & Hutter, 2017), initializing with a learning rate of 1e-4 and a weight decay of 5e-2. Training is conducted over 160k iterations, with learning rate reductions scheduled at the 112k mark. We process five frames from each video in a batch of eight during training, adjusting the frame sizes to maintain a shorter side between 320 and 640 pixels, and ensuring the longer side does not exceed 768 pixels. In all experimental settings, we incorporate COCO joint training, as utilized in prior works (Wu et al., 2022a; Heo et al., 2022; 2023; Ying et al., 2023; Zhang et al., 2023a). For inference, the shorter side of input frames is scaled to 480 pixels, maintaining uniform aspect ratios. We empirically set $\lambda_{\text{sim}}$ as 1.0. In the online experiments using the R50 and ViT-L backbones, eight RTX2080 Ti GPUs are employed. For the offline experiments, eight RTX3090 Ti GPUs are used, while the experiments utilizing the ViT-H backbone are conducted with eight RTXA6000 GPUs.

**Segmentation network.** To achieve distinctive object representation, we employ the following contrastive loss for pretraining the segmentation network $\mathcal{S}$:

$$\mathcal{L}_{\text{embed}} = -\log \frac{\exp\left(\mathbf{v} \cdot \mathbf{k}^+\right)}{\exp\left(\mathbf{v} \cdot \mathbf{k}^+\right) + \sum_{\mathbf{k}^-} \exp\left(\mathbf{v} \cdot \mathbf{k}^-\right)} = \log\left[1 + \sum_{\mathbf{k}^-} \exp\left(\mathbf{v} \cdot \mathbf{k}^- - \mathbf{v} \cdot \mathbf{k}^+\right)\right], \tag{13}$$

where $\mathbf{k}^+$, and $\mathbf{k}^-$ denote positive embedding and negative embedding from anchor embedding $\mathbf{v}$. This contrastive loss is widely applied in the VIS field (Wu et al., 2022b; Li et al., 2023b; Ying et al., 2023; Zhang et al., 2023b; Lee et al., 2024), learning frame-to-frame associations to create better object representations.

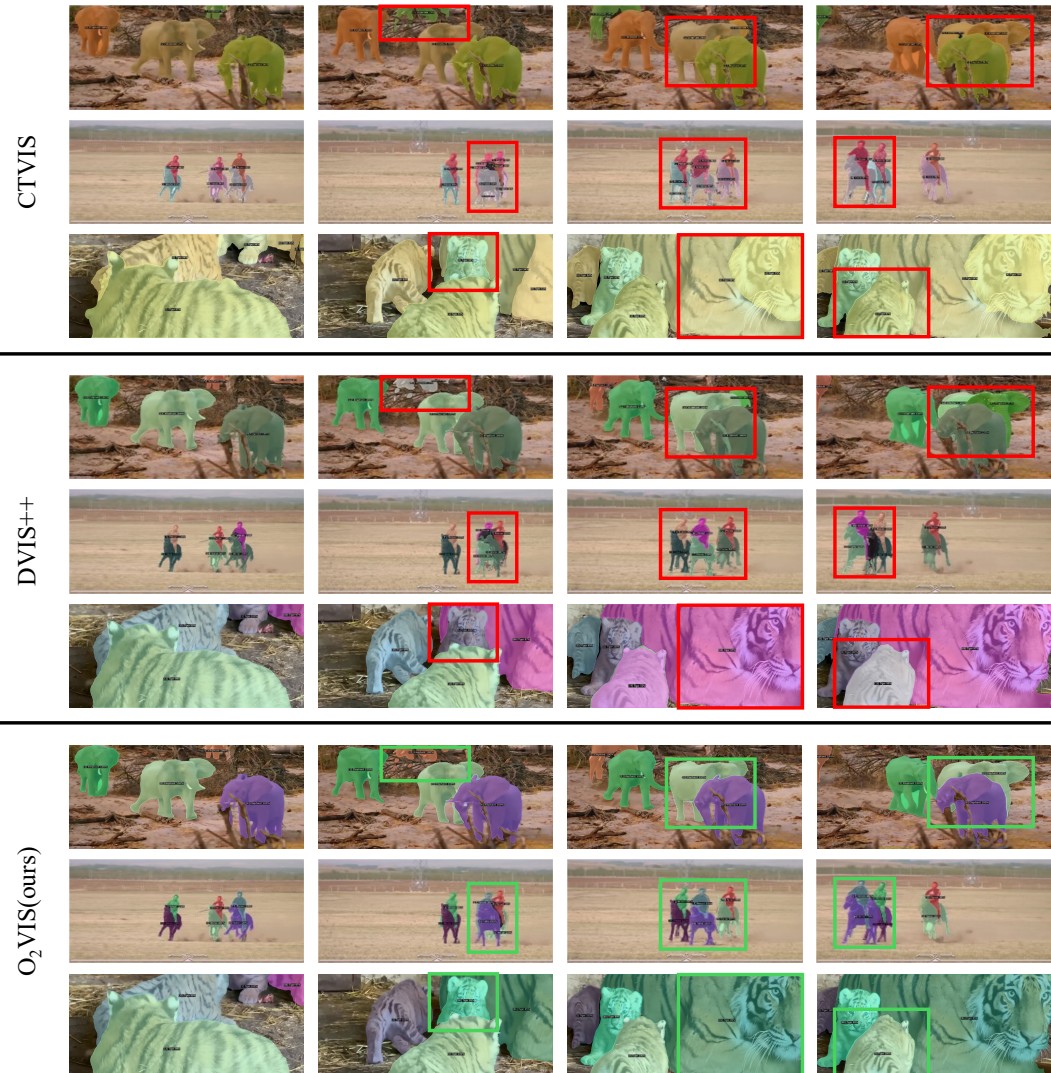

Figure 7: Results of O$_2$VIS, DVIS++ and CTVIS on OVIS dataset

**Early training.** The initial outputs from the tracking networks $\mathcal{T}_E$ and $\mathcal{T}_A$ are also typically noisy. To address this, we utilize the predictions $\hat{y}$ from $\tilde{Q}_t^*$ for ground truth assignment, formulated as:

$$\acute{\sigma} = \arg\min_{\sigma \in \mathfrak{S}_N} \sum_{n=1}^{N_{GT}} \mathcal{L}_{\text{Match}}\left(y_{f(n)}^n, \hat{y}_{f(n)}^{\sigma(n)}\right). \tag{14}$$

The prediction $\hat{y}$ provides guidance for rapid convergence in the same format as the tracked output of MinVIS (Huang et al., 2022).

### A.3 ADDITIONAL QUALITATIVE RESULTS

We provide additional comparisons with state-of-the-art models as shown in Fig. 7 to demonstrate the robustness of our model in various scenarios. DVIS++ (Zhang et al., 2023b) faces challenges with occlusions and rapid motions, resulting in less distinctive embeddings. Similarly, CTVIS (Ying et al., 2023) has difficulty with objects of similar categories, which affects the distinctiveness of embeddings. By employing a decoupled object association strategy, our model ensures consistently discriminative embeddings, thereby improving segmentation and tracking performance.

