# OpenReview forum: "$\text{O}_\text{2}$VIS: Occupancy-aware Object Association for Temporally Consistent Video Instance Segmentation"
_ICLR.cc/2025/Conference — ICLR 2025 Conference Withdrawn Submission_

### Official Review · Reviewer_s7AP · 2024-11-02

**Soundness:** 3
**Presentation:** 3
**Contribution:** 2
**Rating:** 3
**Confidence:** 4

**Summary:**

This paper introduces an occupancy memory and a decoupled object association to track global features of objects long term and to ensure consistent matching of new and old objects. The proposed method achieves good performance on the VIS task.

**Strengths:**

The paper is well-written and easy to follow. A thorough experimental analysis has been performed.

**Weaknesses:**

1. Limited technical novelty: The paper proposes some techniques to improve VIS, but all of these techniques have been seen in other tracking/VIS works in some form or the other. For example, the IOM is similar to global track queries in trackformer [2], Hungarian matching to align current objects to a global memory in DOA has been explored before in many works.
2. Incremental improvement: The results in table 1 and 2 often show a minimal improvement as compared to prior works. For example, on the youtube vis 2019 dataset, the method only gets a 0.2 points improvement over DVIS++ using the R50 backbone. Similar trend is observed for other datasets and other backbones. These improvements could often just come from randomness during training, so it would be nice if the authors could put error bars in the tables to demonstrate consistency.
3. Some prior works (e.g., CAROQ [1]) use query-based propagation for global tracking. How does the proposed method compare with such a method in terms of the number of parameters involved in tracking and the tracking speed? The proposed method requires 2 networks for tracking, as opposed to 1 network in most prior works, so some comparison table on the average time taken and the parameters involved solely for tracking would also be insightful.
4. There are some typos in the paper, e.g., a capitalized letter mid-sentence in line 47.


[1] Choudhuri et al., Context-Aware Relative Object Queries to Unify Video Instance and Panoptic Segmentation, CVPR 2023
[2] TrackFormer: Multi-Object Tracking with Transformers, Meinhardt et al., CVPR 2022

**Questions:**

Please see Weaknesses.

---

### Official Review · Reviewer_reMk · 2024-11-03

**Soundness:** 3
**Presentation:** 3
**Contribution:** 3
**Rating:** 6
**Confidence:** 3

**Summary:**

This paper introduces O2VIS, a novel framework designed to enhance long-term consistency in video instance segmentation. The work presents two main technical innovations: an Instance Occupancy Memory (IOM) for tracking global instance features and their occupancy status, and a Decoupled Object Association (DOA) strategy that separately handles existing and new objects. The framework demonstrates state-of-the-art performance on YouTube-VIS benchmarks.

**Strengths:**

1 The paper's technical contributions are both novel and well-executed. The IOM mechanism provides an elegant solution to the challenging problem of maintaining object identity consistency, while the decoupled association strategy effectively addresses the issue of new object appearances.

2 The comprehensive experimental evaluation, including extensive ablation studies, convincingly demonstrates the effectiveness of each component. The strong performance across multiple benchmarks further validates the proposed approach.

**Weaknesses:**

1 The paper does not provide comparisons of model parameters and inference speed with existing methods, making it difficult to assess the practical implications of implementing this approach.

2 There is no discussion of memory consumption or runtime benchmarks, which are crucial considerations for real-world applications.

3 Some technical details, particularly regarding the IOM update mechanism and the interaction between TE and TA trackers, could be explained more thoroughly.

**Questions:**

1 The authors should include a comprehensive comparison of computational resources, including model parameters, inference speed, and memory usage, with existing methods. This would provide crucial context for understanding the practical trade-offs of their approach.

2 Additionally, including more detailed pseudo-code for key algorithms and visualizations of memory usage patterns would enhance the technical clarity of the paper.

3 Finally, an analysis of failure cases and performance on longer video sequences would provide valuable insights into the method's limitations and potential areas for improvement.

---

### Official Review · Reviewer_nofb · 2024-11-03

**Soundness:** 2
**Presentation:** 2
**Contribution:** 2
**Rating:** 6
**Confidence:** 2

**Summary:**

This paper introduces O2VIS, a novel framework for video instance segmentation that enhances long-term consistency in object tracking. The framework incorporates an Instance Occupancy Memory (IOM) module and a Decoupled Object Association (DOA) strategy, effectively distinguishing between new and recurring objects across frames. By decoupling the association of existing and newly appeared objects, the method maintains stable and consistent object identities throughout videos. Experimental results demonstrate that O2VIS achieves state-of-the-art AP scores on the YouTube-VIS 2019, 2021, and 2022 datasets, setting a new benchmark for the VIS task.

**Strengths:**

Innovation: The study proposes an instance occupancy memory mechanism that addresses challenges in maintaining consistency when objects disappear and reappear, making it well-suited for complex, dynamic video scenes.
Performance: The experimental results show that O2VIS significantly outperforms current state-of-the-art methods across multiple datasets, especially in AP scores.
Decoupled Strategy: By implementing a decoupled association strategy for handling existing and new objects separately, the method avoids common background misalignment issues, enhancing tracking accuracy.
Comprehensive Experiments: The paper provides thorough experimental comparisons with existing VIS methods, and ablation studies validate the effectiveness of each technical component, demonstrating the contributions of IOM and DOA modules.

**Weaknesses:**

Insufficient Details: While the paper introduces the Occupancy-guided Hungarian Matching and Decoupled Object Association strategies, implementation details are limited. Providing pseudocode or more concrete algorithmic descriptions could enhance clarity.
Computational Cost: The addition of IOM and DOA likely increases computational complexity, particularly due to multi-frame memory updates and association matching. It would be beneficial to quantify the computational overhead of these modules within the paper.
Generalizability: Experiments are currently focused on standard datasets like YouTube-VIS. The model’s performance in more challenging scenarios, such as high occlusion or rapid object movement, remains unclear.
Model Complexity: With the integration of multiple modules, the overall model structure is complex, which may pose deployment challenges. Future work could explore simplifying the model or improving its efficiency.

**Questions:**

see Weaknesses

---

### Official Review · Reviewer_kMha · 2024-11-03

**Soundness:** 2
**Presentation:** 1
**Contribution:** 2
**Rating:** 3
**Confidence:** 3

**Summary:**

The paper presents a method for long-term tracking consistency in videos for the task of video instance segmentation. The core idea is that using the visibility or occupancy of the objects can help in associating their features correctly so as to differentiate between new and previously seen objects. Treating these two kinds of objects differently by associating them separately also helps. Experiments exist that compare state-of-the-art approaches to the proposed method.

**Strengths:**

- The motivation is well-grounded that the objects should be treated differently based on if they have been seen before

**Weaknesses:**

- One of the most important baselines is missing for this task is SAM2 for video instance segmentation
- While the motivation is good, similar ideas have appeared before for tracking, for instance in DeepSORT or Detecting Invisible People. These are not segmentation approaches but should be cited to acknowledge that both of the main contributions of this paper have appeared before.
- It seems like the ID switches metric from multi-object tracking based on bounding boxes, is what the paper wanted to improve but there is no comparison to prior approaches with that metric so it is hard to tell if their claim of long-term consistency is valid over an entire dataset.

**Questions:**

- It is very hard to understand Figure 1. There are barely any labels and barely any text in the caption to explain what each of the icons in the figure means. The first teaser figure should be very easy to understand and should convey an overall takeaway from the method, and not describe the method itself.
- Can you explain how you get an object's occupancy O near L206?
- If an object's occupancy is 0, why should it's new feature representation be added to the memory?

---

### Official Review · Reviewer_yELW · 2024-11-04

**Soundness:** 2
**Presentation:** 2
**Contribution:** 2
**Rating:** 3
**Confidence:** 4

**Summary:**

This paper presents the O2VIS framework, aiming to improve long-term consistency in video instance segmentation. By introducing Instance Occupancy Memory (IOM) and Decoupled Object Association (DOA), this method enhances the stability of object tracking in dynamic video scenes, effectively differentiating between recurring and new objects. The paper demonstrates the approach's performance on multiple benchmark datasets, such as YouTube-VIS and OVIS, highlighting its advantages in accuracy.

**Strengths:**

1. Application-oriented Innovation: O2VIS introduces occupancy information into the memory update process, making it possible to maintain object identity consistency more accurately in scenes where objects frequently appear and disappear. This occupancy-aware memory management strategy provides a useful enhancement for video instance segmentation.

2. Empirical Support: The experimental results on various benchmark datasets show improved average precision (AP) and average recall (AR), supporting the method's effectiveness. Additionally, the ablation studies validate the contributions of IOM and DOA, strengthening the reliability of the results.

3. Well-designed Component Structure: The decoupled approach in DOA separately manages existing and new objects, using occupancy-guided Hungarian matching to reduce incorrect associations. This is a practical and effective design choice.

**Weaknesses:**

1. Limited Theoretical Innovation: The calculation of foreground probability is essentially a weighted adjustment using output probabilities and does not introduce a novel computational framework or algorithm. IOM and DOA represent more of an applied enhancement of existing memory and association techniques, rather than a fundamental theoretical breakthrough, which may limit the impact at conferences focused on theoretical innovation.

2. Unclear Generalizability: The method is primarily designed for video instance segmentation, and its applicability to other tasks, such as multi-object tracking, has not been demonstrated. Verifying IOM and DOA’s effectiveness in other visual tasks would strengthen the paper’s generalizability.

3. Dependence on Pre-trained Model Accuracy: Since the foreground probability relies on classification outputs, errors in these outputs could lead to incorrect memory updates, potentially destabilizing tracking performance. This dependency might reduce overall system stability, particularly when applied to longer or more complex video sequences.

**Questions:**

Is the effectiveness of IOM and DOA useful in other tasks such as multiple object tracking?

---

### Note · Authors · 2024-11-13

I have read and agree with the venue's withdrawal policy on behalf of myself and my co-authors.